# From Probiotics to Psychobiotics: Live Beneficial Bacteria Which Act on the Brain-Gut Axis

**DOI:** 10.3390/nu11040890

**Published:** 2019-04-20

**Authors:** Luis G. Bermúdez-Humarán, Eva Salinas, Genaro G. Ortiz, Luis J. Ramirez-Jirano, J. Alejandro Morales, Oscar K. Bitzer-Quintero

**Affiliations:** 1INRA, Micalis Institute, INRA, AgroParisTech, Université Paris-Saclay, 78352 Jouy-en-Josas, France; 2Department of Microbiology, Center of Basic Science, Universidad Autónoma de Aguascalientes, Aguascalientes 20131, Mexico; emsalin@correo.uaa.mx; 3Neurosciences Division, Centro de Investigación Biomédica de Occidente, Instituto Mexicano del Seguro Social, Guadalajara 44340, Jalisco, Mexico; genarogabriel@yahoo.com (G.G.O.); ramirez_jirano@hotmail.com (L.J.R.-J.); 4Department of Computer Sciences, Universidad de Guadalajara, Guadalajara 44430, Jalisco, Mexico; jalejandro.morales@academicos.udg.mx

**Keywords:** probiotics, microbiota, beneficial bacteria, psychobiotics, human health

## Abstract

There is an important relationship between probiotics, psychobiotics and cognitive and behavioral processes, which include neurological, metabolic, hormonal and immunological signaling pathways; the alteration in these systems may cause alterations in behavior (mood) and cognitive level (learning and memory). Psychobiotics have been considered key elements in affective disorders and the immune system, in addition to their effect encompassing the regulation of neuroimmune regulation and control axes (the hypothalamic-pituitary-adrenal axis or HPA, the sympathetic-adrenal-medullary axis or SAM and the inflammatory reflex) in diseases of the nervous system. The aim of this review is to summarize the recent findings about psychobiotics, the brain-gut axis and the immune system. The review focuses on a very new and interesting field that relates the microbiota of the intestine with diseases of the nervous system and its possible treatment, in neuroimmunomodulation area. Indeed, although probiotic bacteria will be concentrated after ingestion, mainly in the intestinal epithelium (where they provide the host with essential nutrients and modulation of the immune system), they may also produce neuroactive substances which act on the brain-gut axis.

## 1. Introduction

The skin and mucosal surfaces of vertebrates contain a wide collection of microorganisms (collectively named microbiota) which includes bacteria, fungi, parasites and viruses. The human gut harbors one of the most complex and abundant ecosystems composed of up to 10^13^–10^14^ microorganisms which is between 1 to 10 times more than the number of eukaryotic cells in the body [1,2]. The collective adult human gut microbiota is composed of a maximum of 500–1000 bacterial species [1,3,4].

Hundreds of years of co-evolution have led to a mutual symbiosis between the host and gut microbiome. Indeed, the gut is rich in molecules that can be used as nutrients by the microorganisms, favoring microbiota colonization [1]. Gut colonization begins at birth and is established in the first 3 years of life. The initial interaction between gut microbiota and the host is indispensable for the maturation of the nervous system, the immune system and for the developmental regulation of intestinal physiology [1,5,6]. At this stage, gut microbiota is also able to modulate the process of angiogenesis [7]. Furthermore, microorganisms also display anti-microbial activities, thus maintaining a stable gut ecosystem. Alterations in the process of microbial colonization of the human gut in early life have been shown to influence the risk of disease [8].

Later in life, microbial colonization of the intestine has a significant impact on the host neurophysiology, behavior and function of the nervous system [9,10,11]. Given the immunomodulatory properties of gut microbiota, it has been shown that different immune pathways, inside and outside the central nervous system (CNS) are involved in important mechanisms like microbial mediation of brain functions and behavior. It has been discovered that neuroimmune modulation by the microbiota is able to contribute to etio-pathogenesis or to display important signs and symptoms in neurodegenerative and behavioral disorders such as autism spectrum disorders (ASD), anxiety, depression, Alzheimer’s disease (AD) and Parkinson’s disease (PD) [9].

Furthermore, different bacteria, which are commonly present in a large diversity of food products, transit through our gut every day, interacting with the food products themselves, the host microbiota, and our own cells in either a healthy or a pathological context. Many of the latter microorganisms are known as probiotics. Probiotics are defined as “live microorganisms which when administered in adequate amounts confer a health benefit on the host” [12]. This concept is based on the observations made by Élie Metchnikoff in 1907 in which the regular consumption of lactic acid bacteria (LAB) in fermented dairy products, such as yogurt, was associated with enhanced health and longevity in many people living in Bulgarian villages.

In association with probiotics, the concept of prebiotic was firstly introduced by Gibson and Roberfroid in 1995 as a “non­digestible food ingredient that beneficially affects the host by selectively stimulating the growth and/or activity of one or a limited number of bacteria already resident in the colon” [13]. This definition implies their ability to resist host digestion and their unique activity on microbiota intestinal. However, the progress in prebiotic study has forced the evolution of this definition, as they can also be directly administered to other sites of the body that are also colonized by commensal bacteria, such as skin or vagina. Recently, the International Scientific Association for Probiotics and Prebiotics (ISAPP) in a consensus panel proposed a new definition of prebiotics as “a substrate that is selectively utilized by host microorganisms conferring a health benefit”, considering that dietary ones must not be degraded by host enzymes [14]. Substrates that fit with this definition include oligosaccharides (OS), polyunsaturated fatty acid, conjugated linoleic acid, plant polyphenols, and certain fermentable fibers. Among prebiotics highlight fructooligosaccharides (FOS), inulin, galactooligosaccharides (GOS), mannanoligosaccharides (MOS), xylooligosaccharides (XOS) and human milk oligosaccharides (HMO), being FOS and GOS the most studied as well as the classically accepted ones [14]. Prebiotics have been demonstrated to exert beneficial effects on the gastrointestinal tract, immune system, bones, lipid and sugar metabolism, and on mental health.

Dinan and coworkers [15] originally defined psychobiotics as probiotics that, upon ingestion in adequate amounts, yield positive influence on mental health. Because prebiotics have demonstrated benefit on mental health and they support the growing of specific commensal bacteria with psychophysiological effects, prebiotics can be included in the definition of psychobiotics [16]. In this sense, most prebiotic substrates analyzed for their neural effects are FOS and GOS, which favorably stimulate the growth of bifidobacteria and lactobacilli.

The effect of psychobiotics is not limited to the regulation of the neuroimmune axes (hypothalamic pituitary adrenal (HPA)-axis, sympatho-adrenal medullary (SAM)-axis and the inflammatory reflex) and in diseases that involve the nervous system, but they are also related to cognitive, memory, learning and behavior. Psychobiotics have thus opened a very broad and interesting panorama that changes the current paradigm of symbiosis between bacteria and humans. From this new point of view, this relationship seems to be more a commensalism, rather than a pure symbiosis.

## 2. Axes of Neuroimmune Control and Regulation

It has been shown that there is an important neural control of the immune system [17]. A well-known principle of the physiology in mammals is that the nervous system is responsible for achieving homeostasis by modulating of the function of other systems in the body through the HPA axis, the inflammatory reflex, the enteric nervous system (ENS) and finally the brain-gut axis.

Microglia is the resident immune cell in the CNS which represents 5 to 20% of glial cells. It is a myeloid cell, phagocytic, and has the activity of an antigen presenting cell (APC). In addition, it releases cytokines and can activate inflammatory-type responses [18,19]. During the early development stage, the microglia “brand” and “clean” synapses through a process called “synaptic pruning”, promotes the “wiring” of neuronal circuits and releases cytokines and chemokines that assist and guide the process of neuronal differentiation [18,20]. The microbiota has a direct influence on the maturation and function of the microglia. In germ-free (GF)-animals, the microglia display a longer development process and with more derivations, with high levels in the expression of receptor-1 of the colony stimulating factor (CSF1R), F4/80 and CD31, factors that decrease in expression during development. This suggests that there is an important effect of the microbiota on the microglia, which depends on the stage of development and/or the time of microbial colonization.

The microglia of adult GF-mice can be functionally damaged when there are alterations caused by lipopolysaccharide (LPS) or by lymphocytic choriomeningitis virus, which in turn causes alterations in the correct activation of the immune system, including an increase in the release of proinflammatory cytokines such as tumor necrosis factor (TNF)-α, interleukin (IL)-1β and IL-6. These functional deficits are consistent with the concept that the naive microglia of adult GF-mice has a significant decrease in the expression of several genes important to interferon (IFN)-mediated responses, in genes for innate immune responses and genes for viral defense response and effector processes [9,21].

The mechanisms through which intestinal microbes exert their influence on microglia in the brain are not clear but it seems that there is a “microglial modulation” according to a specific type of bacteria [9]. This has raised the question of whether the effects of microbiota on microglia are not regulated by bacteria in general, by the microbiome, or if very specific microbial species are required [9]. The alterations in the morphology of the microglia in GF-animals and the alteration in the expression of genes can be normalized thanks to the post-natal supplementation with short-chain fatty acids (SCFAs), which are products of bacterial fermentation [22,23], suggesting that the bacterial species producing SCFAs are able to restore the alterations that occur in the microglia in GF-mice or treated with antibiotics [9].

The coordination of information between neurons, microglia and the responses at the central level with the periphery is carried out through the different axes of regulation and control; the HPA axis, and the inflammatory reflex (Figure 1). The coordination of these defense responses is mediated by signaling pathways related to the hypothalamus, the pituitary gland and the adrenal glands (e.g., HPA-axis), which causes the release of chemical molecules capable of altering behavior, including glucocorticoids, mineralocorticoids, and catecholamines. The activity of the HPA-axis is regulated by multiple sympathetic, parasympathetic and limbic circuits (amygdala, hippocampus and medial prefrontal cortex) that will directly or indirectly activate the hypothalamic paraventricular nucleus (PVN) [24]. Under normal conditions HPA-axis activity exhibits continuous oscillatory activity synchronized with circadian as well as ultradian rhythms [25,26].

The sympathic nervous system (SNS) and HPA-axis activation are the main components of neurotransmitter release and of neuroendocrine molecules of the stress response [27]. To respond to stress, the SNS is responsible for increasing catecholamine levels in the systemic circulation and tissues, with the concomitant release of corticotropin releasing factor (CRF) from hypothalamic paraventricular neurons, then the release of the adrenocorticotropic hormone (ACTH) from the anterior pituitary gland is stimulated, and ACTH travels through the circulation systemic and induces the synthesis and release of glucocorticoids from adrenals, cortisol in humans and corticosterone in animals. The primary function of the SNS and HPA-axis activation is to prepare the body to respond to damage by increasing the level of glucose in the blood through gluconeogenesis, the suppression of the immune system (suppression of cytokines) and increasing the metabolism of fats and proteins [27,28].

The responses of the HPA-axis as well as other variables of the stress response are regulated by exposure to psychological and physical stressors, such as infections [29]. The response of the HPA-axis and the SAM axis to psychological stress is mediated by neurotransmitter systems such as serotonin (5-HT), norepinepherine (NE) and endorphins, which play an important inhibitory role [29].

In addition, the HPA axis is strongly regulated to react efficiently to pathogens such as *Escherichia coli*. This response is mediated by the synthesis of prostanoids induced by the enzyme cyclo-oxygenase (COX). The elevation in corticosterone levels correlates with the increase in prostaglandin (PG) E2 in the circulation [29].

Interestingly, prebiotic intake in early life has been associated with beneficial neurological effects in adulthood. To demonstrate this effect, William and co-workers [30] fed neonatal rats (3 days old) with B-GOS during 19 or 53 days and showed that levels of the *N*-methyl-d-aspartate receptor (NMDAR) N2A subunit, synaptophysin and the brain-derived neurotrophic factor (BDNF) in the hippocampus of adult rats were elevated as compared with control fed animals. As the expression of the microtubule-associated-protein-2 (MAP2) was not affected, authors propose that neonatal B-GOS feeding impacts on neurotransmission, but not on synaptic architecture. Similar results were reported by Oliveros group [31] supplementing rat pups with 2′-fucosyllactose (FL) during the lactation period. When animals were evaluated just after weaning there was no change in behavior, although 2’-FL-feed rats evoked more intense long-term potentiation (LTP) than control ones. Same animals were evaluated at the age of 1 year and they performed significantly better in behavioral tests and still evoked more intense and longer TLP than the control group. Taken together, these results show that prebiotic administration early in life improves cognitive abilities both in childhood and in adulthood. However, in a randomized controlled trial, no significant improvement in neurodevelopmental outcomes was observed in preterm infants fed with breast milk or preterm formula supplemented with short-chain GOS/long-chain FOS/pectin-derived acidic OS between days 3 and 30 of life when they were evaluated at one or two years of age [32,33].

The basic organizational unit of the nervous system is the reflex arc, which is composed of sensory neurons (afferent) that report information to the CNS and motor neurons (efferent) that send “regulatory” signals to “target” tissues in the periphery. Recent advances in both neuroscience and immunology have revealed that there are neural reflexes that can regulate immune function in a wide range of species through evolution, from *Caenorhabditis elegans* to more complex mammals [17,34,35].

Inflammation is a key process of mammals in order to fight against pathogenic microorganisms and in the mechanism of wound healing. The molecular products of a bacterial invasion and of damaged tissue are rapidly detected by pattern recognition receptors (PRRs), which activate the cells of the innate immune system. The early response of these cells starts a cascade of events whose main function is the exclusion of pathogens and the subsequent restoration of homeostasis. This process includes the synthesis and release of proinflammatory cytokines and leukocyte recruitment [17].

It is crucial for the host to regulate and control an inflammatory response. Different mechanisms of regulation and control of inflammatory mediators have been described; for example, the release of inhibitory cytokines and soluble receptors to cytokines, as well as the activation of different subtypes of regulatory lymphocytes [36]. It is interesting to note that PRRs (Toll-like receptors, TLRs, and Nucleotide-binding and oligomerization domain (NOD)-like receptors, NLRs), in addition to the receptors for cytokines and PG are also expressed by sensory neurons [37]. This provides a molecular mechanism by which the CNS acquires information from a process of inflammation localized in the periphery. In addition, sensory nerves can react to the presence of microbial products independently of the activation of the immune system [17]. These nerves form a dense network along the external surfaces of the organism, and it has been suggested that this type of innervation provides the anatomical basis for a very precise “sensing” by the CNS against a potential infection with a pathogen, a tissue damage or an inflammatory process [17,37]. Reciprocally, many of the cells of the immune system express receptors for neurotransmitters such as dopamine (DA), acetylcholine (Ach) and norepinephrine (NE), all of which in turn regulate the differentiation and activity of leukocytes [17,38,39,40].

In this phenomenon, afferent signals are transmitted through the “vagus nerve”, which are processed at the central level (CNS) and return to the periphery via the vagus efferent nerve, a process by which the release of cytokines is regulated by splenic macrophages [17,41]. Spleen is the major organ where TNF-α is synthesized and released systematically during an endotoxic process (endotoxemia). It has been shown that electrical stimulation of the vagus nerve significantly reduces the release of TNF-α in the spleen. However, the vagus nerve does not directly supply the spleen, the signal travels to the celiac ganglion, where the splenic adrenergic nerve also flows. The electrical stimulation of the latter also reduces the synthesis of TNF-α in the spleen. For this inhibition to occur, activation of the α7 subunit of the nicotinic acetylcholine receptor (α7nAchR) in splenic macrophages is necessary. The adrenergic nerve terminals are very close to a sub-type of T lymphocytes in the spleen that expresses the enzyme choline acetyltransferase (AchT), which has the ability to synthesize Ach, a neurotransmitter that is necessary to inhibit the synthesis and release of TNF-α in the spleen [42].

The vagus nerve, in addition to the splenic and splanchnic nerves, provide an important line of communication with the HPA-axis [29]. For instances, 2 h after vagal stimulation in rodents there is an increase in the expression of the mRNA of CRF in the hypothalamus and corticotropin-releasing hormone (CRH) which in turn increases the levels of ACTH and plasma corticosterone levels. The clinical relevance is the fact that vagal stimulation is associated with clinical benefits (signs and symptoms) antidepressants, together with the “normalization” of HPA-axis parameters in patients with refractory depression [43].

The impact of other prebiotics on brain physiology and biochemistry has also been experimentally studied. Rats that received a diet for 5 weeks supplemented with 2’-FL, the most abundant HMO, experienced increased BNDF levels in the striatum and hippocampus [44]. The expression of other two brain functional markers involved in the LTP process, the postsynaptic density protein (PSD)-95 and phosphorylated calcium/calmodulin-dependent kinase II (pCaMKII), was also augmented at frontal cortex and hippocampus, and at hippocampus, respectively. In accordance, authors reported an enhancement of synaptic plasticity in rats with that feeding regimen and in mice with a 2’-FL long-term feeding (12 weeks) protocol. Both species improved input/output curves and LPT experimentally evoked at hippocampal synapses, with a better performance of the animals in different applied tests of learning behavior.

Finally, some studies have addressed the impact of prebiotics in experimental models of neural dysfunctions. The first one was focused on analyzing the effect of GOS intake in a mouse model of amyotrophic lateral sclerosis (ALS) [45]. Animals started to receive the prebiotic at the age of 8 weeks on a daily basis until the end of the protocol. Mice orally fed with GOS experienced delayed onset of the disease, extended lifespan, improved muscle atrophy, attenuated oxidative stress of skeletal muscles, suppressed astrocyte and microglia activation, inflammatory response and apoptosis in spinal cord tissue. Authors attributed neuroprotective effects of GOS on ALS-sick mice to the amelioration on homocysteine serum levels, an amino acid related to neurotoxic effects in the pathogenesis of ALS, and to the increases in the amount of VitB12 and folato, both of which are involved in homocysteine metabolism [46]. Besides, beneficial effects of GOS have been also described in neuropsychiatric disorders where anxiety and neuroinflammation are clinically involved [47]. In this sense, the supplementation of 8 weeks-old mice standard diet with B-GOS incorporated to drink water during 3 weeks reduced LPS-induced anxiety. B-GOS intake also decreased elevated cortical IL-1β and 5-HT2A receptor expression mediated by LPS in the frontal cortex, in the absence of altered 5-HT metabolism. Thus, the anti-inflammatory effect of the prebiotic is probably modulating its anxiolytic activity. A similar link between anti-inflammation and neuroprotection was reported for chitosan oligosaccharides (CHO) [48] and FOS [49] in a rat model of AD. When CHO was orally administered to amyloid-β_1-42_-induced rats during 2 weeks, the learning and memory deficits were reduced, and the hippocampal cell death decreased. At the same time, CHO treatment inhibited oxidative stress together with a reduction in proinflammatory cytokines expression at the hippocampus, particularly IL-1β and TNF-α [48]. In the case of a study done with FOS, it was orally and daily administered to amyloid-β_1-42_-induced rats for 4 weeks or to D-Galactose-induced rats for 8 weeks, with similar outcomes. FOS intake improves inflammation and oxidative stress disorder, ameliorates learning and memory difficulties, and regulates the synthesis and secretion of neurotransmitters such as NE, DA, 5-HT, and 5-hydroxyindole acetic acid (5-HIAA) [49]. All these FOS-induced effects on AD are mediated by the regulation of the gut microbiota.

## 3. The Interaction of Microbiota with Enteric Nervous System and Brain-Gut Axes

In the last 10 years the importance of the brain-gut axis has been highlighted [50,51,52]. A connection has been established between the gut and the CNS, which is essential to achieve host homeostasis. It has been called the “brain-gut axis” or “GB axis” [53] (Figure 2). The brain-gut axis includes: the CNS, neuroendocrine and neuroimmune systems, the sympathetic and parasympathetic “arms” of the autonomic nervous system (ANS), the enteric nervous system (ENS) and noticeably the intestinal microbiota [29]. All these components interact and form a very complex network of reflexes, with afferent fibers (input) that project towards integrative structures of the CNS and efferent fibers (output) with projections towards the smooth muscle. This bi-directional communication network enables the sending of signals from the brain and influences the motor, sensory and secretory part of the gut, and conversely, visceral messages from the intestine can influence brain functions, especially in areas dedicated to the regulation of stress at the hypothalamic level [29].

The sympathetic nervous system (SNP) enables the selective presentation of enteric bacteria to the mucosal immune system. Nerve fibers containing NE have been identified very close to the epithelium surrounding the lymphoid follicles in the jejunum of pigs; the administration of NE increases the reception of pathogenic bacteria inside the follicles [7]. In this sense, it has been suggested that the release of biogenic amines, such as NE, can influence the composition of the intestinal microbiota. For instance, it has been observed that this neurotransmitter stimulates the growth of both pathogenic and nonpathogenic *Escherichia coli in vitro*, in addition to influencing its adherence to the mucous membranes [48,54,55]. Changes in the physiology of the host that originated within the gut or from signals from the CNS, produce changes in the bacterial composition of the gut [7].

The ENS is a complex neuronal network that involves multiple neurotransmitters such as 5-HT, Ach and CRF, where a prominent role is given to the CRF that is mediating changes at the level of gastrointestinal function. At the ENS level, this CRF demonstrates that peripheral pathways also play a preponderant role in the local regulation of the intestine and its function in states of stress [56]. The activation of CRF-1 receptor (CRFR1) in the intestine induced by stress increases the motility of the colon, defecation, permeability of the intestine and the sensation of visceral pain [57]. The activation of CRFR2 inhibits gastric emptying, suppresses the motor function of stimulating the colon and prevents hypersensitivity generated by colorectal distension. It has been proposed that CRFR2 may have a key role in stress-induced patency dysfunction and in mucosal immune modulation and in inflammatory responses in the colon [58]. CRF can directly activate mesenteric neurons to increase motility, permeability and stimulate diarrhea in rodents [59].

The contrasting actions of CRFR1 and CRFR2 are associated with differential expression patterns. CRFR2 is present in anterior regions of the intestinal tract [60], whereas CRFR1 is mostly distributed in the colon, and is expressed in a very important way in the cells of the colon mucosa [56]. The presence of CRFR2 in the colonic mucus has been demonstrated and it has been proposed that in this area it may also have an important role in the stress-induced patency dysfunction in the modulation of immune and inflammatory responses within the colon mucosa [61]. The evidence shows that stress causes the recruitment and activation of CRF receptors in the colon, which induces changes related to the same stress in the intestinal function and in turn causes an increase in sensitivity to stress that results in an altered expression of receptors to CRF [29].

5-HT is recognized as the most important biological substrate in the pathogenesis of mood disorders [62]. There is evidence of the role of serotonergic signaling in the neurobiology of anxiety [63,64]. In GF-mice, altered levels of 5-HT have been reported in the striatum and in the hippocampus, which suggest an association between the microbiota and serotonergic signaling [62]. In addition to its role as a neurotransmitter in the brain, monoamine 5-HT is a potent regulator in the gut. More than 90% of all 5-HT in the body is synthesized in the intestine, where it activates 14 different types of receptors located in enterocytes [65,66], in enteric neurons [67] and in cells of the immune system [68]. In addition, circulating platelets sequester 5-HT from the gut and release it for the purpose of distributing it in different parts of the body [69]. 5-HT derived from the intestine regulates various functions including motor and secretory reflexes, platelet aggregation, regulation of immune responses, bone development, and cardiac function [69]. A dysregulation of peripheral 5-HT levels is implicated in the pathogenesis of diseases such as irritable bowel syndrome (IBS), cardiovascular diseases [70] and in osteoporosis processes. The molecular mechanisms that control the metabolism of 5-HT at the intestinal level are still unclear, but it has been shown to be synthesized by specialized endocrine cells called enterochromaffin cells (ECs), as well as by mast cells of the mucosa and by mesenteric neurons (Figure 3) [69].

Exposure to chronic psychosocial stress decreases the levels of *Bacteroides* spp. and increases the levels of *Clostridium* spp. in the caecum, while increasing circulating levels of IL-6 and CCL2 chemokine (monocyte chemoattractant protein, MCP-1), which is indicative of an immune activation. The levels of IL-6 and CCL2 correlate with changes in the levels of *Coprococcus* spp., *Pseudobutyrivibrio* spp. and *Dorea* spp. induced by stressors directly in the intestine [53].

Some types of bacteria, such as lactobacilli, are able to convert nitrate to nitric oxide (NO), a potent regulator of responses to different levels of immune and nervous system. Lactobacilli also increase the activity of the enzyme indol-amine-2,3-dioxygenase (IDO), involved in the catabolism of tryptophan (TRP) and in formation of neuroactive compounds of kinuric and quinolinic acid [71]. Modification of the intestinal microbiota in adult mice causes changes in behavior, which may be related to immune, neural and hormonal mechanisms. In relation to immune mechanisms, it is known that TLR-2, 4 and 5 are over-regulated in the gut during colonization, which implies that there is interaction between these receptors and the microbiota [72,73]. The dendritic cells (DCs) of the gut break the epithelial layer and interact with commensal bacteria to induce the production of immunoglobulin-A (IgA) by B-lymphocytes and plasma cells. The secreted IgA confines penetration of the microbiota through the epithelium. This mechanism hampers an inflammatory response to commensal bacteria under normal conditions. The DCs are very close to nerve areas of the gut, the neuropeptide sensor calcitonin gene-related peptide (CGRP) modulates the function of these DCs [74] and can send signals about the presence of commensal bacteria to the brain via the vagus nerve [75]. The vagus nerve plays an important role in signaling the gut to the brain and can be stimulated by bacterial products such as endotoxins or inflammatory cytokines such as IL-1β and TNF-α [75]. The vagal response to stimulation by peripheral inflammatory events is the suppression in the release of pro-inflammatory cytokines from intestinal macrophages mediated by α7nAchR [76,77].

Dai, et al. [78] showed that certain probiotics are able to trigger IL-10 mediated anti-inflammatory responses by downregulating the proinflammatory cytokines TNF-α and IL-6. Both of these proinflammatory cytokines, along with IL-2 and IL-1β, are key participants in depressive states and other affective disorders (78). Several other microbe associated molecular patterns (MAMPs) are able to trigger or block inflammatory responses that are associated with different bacterial genera, e.g., bifidobacteria inhibits TLR activation, preventing the inflammatory response [79,80]. Other MAMPs-TLR interactions include OS and the intestinal epithelium. The inflammatory response is directly responsible for the intestinal barrier permeability, nutrient absorption, and microbiome translocation. That is the case in acute stress, that initiates inflammation and secondary dysbiosis, due to aberrant translocation, where the probiotic *Lactobacillus helveticus R0052* has been shown to be able to restore the intestinal barrier [16,79,81,82].

Colonization with *Bacteroides thetaiotaomicron* induces a 2- to 5-fold increase in the expression of mRNA that codes for the synaptic protein-33 associated with vesicles, which is involved in synaptic neurotransmission. This finding confirms that commensal bacteria can influence nervous system functions [7,50]. The intestinal microbiota is essential for the normal development of the immune and mucosal systems, which are intimately associated with the impact of the same microbiota on brain development and function [50].

## 4. Behavior, Cognition, and Emotion

It has been demonstrated that bi-directional communication exists between the intestine and the brain and that it involves neurological, metabolic, hormonal and immunological signaling pathways; and that disturbance or alteration in these systems can result in altered behavior [83]. A clear example is intestinal inflammation, which has been associated with changes in bowel-brain interactions, as well as a high morbidity between inflammatory bowel disorder and anxiety states (Figure 4) [84].

The role of microbiota has not only focused on the impact it exerts on the brain and central nervous function but also on how it is intimately related to the constitutive modulation of nerve function at the peripheral central level [71].

Stress has been defined as a very complex dynamic condition in which homeostasis or the internal “resting state” is altered or threatened [85,86]. Throughout life all organisms are exposed to factors that exceed the homeostatic threshold, which results in a stress response, which may be physical, psychological or immunological. Evolution has armed most organisms with the necessary biological machinery to mount a defense response to acute stressors and restore the homeostatic balance once the stress or damage has subsided [85].

A significant number of animal studies provide abundant evidence that the medial prefrontal cortex (MPFC) plays an important role in the regulation of stress circuitry [28]. While the ventral part of the MPFC has been augmented with a stimulatory role, the more dorsal part in contrast has been described to possess an activity of HPA-axis inhibition. It has been also described that this negative feedback mechanism is mediated by the inhibition of glucocorticoid receptors (GRs) in the MPFC [28]. The amygdala is a key region in the process of stress responses in addition to being an important target for the inhibitory feedback system by the MPFC [87]. In humans, the MPFC area is involved in the modulation of amygdala activity during emotional conflicts and in the regulation of autonomic and affective responses [28,88].

Stress, particularly in the early stages of life, is one of the major predictors of the onset of major depression disorder (MDD) [89]. Early exposure to stress and MDD is associated with a significant de-regularization of the HPA-axis and the stress/cortisol response system. Exposure to stressors, HPA-axis deregulation, elevated corticosteroid levels and major depression states are related to structural alterations in the hippocampus and amygdala, key regions in the regulation of the HPA-axis [90,91].

In one study of early life maternal separation, a group of male rats were submitted to stress tests [79]. They all showed the typical pattern: poor forced swim performance while the group that was separated also showed records of high IL-6 blood levels, low NE levels in brain and higher expression of CRF gene in the amygdala [92]. By administering *L. rhamnosus R011* plus *L. helveticus R0052*, the rats downregulated their HPA axis and normalized their corticosterone levels [16,92].

Psychobiotics are now considered key elements in affective disorders. In one experiment with mice that were administered with *L. rhamnosus*, they featured lesser signs of anxiety and depression in forced swim and plus elevated maze respectively than their control counterparts, even at the same levels of corticosterone [16,93]. This suggests that the probiotic had a downregulation effect over HPA axis [93]. In the presence of *L. rhamnosus*, mice showed a lower hippocampal expression of the GABA_B1b_ receptor gene and a higher expression of it in the cingulated cortex and limbic regions. Since GABA is the main inhibitory neurotransmitter of the nervous system, it would appear that psycobiotics are able to modulate the local balance of inhibition/exciting in order to control the systemic responses to stress, anxiety and depression [93].

As previously described, GF-mice exhibit an exaggerated response to stressors, with the presentation of anxious-type behaviors and cognitive deficits [94,95]. This behavior is influenced by the amygdala and the hippocampus. The signaling between the basolateral amygdala (BLA) and the ventral hippocampus modulates anxiety behaviors and social behaviors [96]. Tune changes (structural changes) in the amygdala and hippocampus are associated with anxiety disorders in humans and in rodents in early stages of development. There is evidence of hypertrophy of the dendrites of excitatory neurons in the BLA area under a state of repeated (repetitive) stress that induces atrophy of the dendrites in hippocampal neurons [94].

The “germ-free” status induces dendritic hypertrophy in inhibitory interneurons, and the excitatory pyramidal neurons of the BLA area show increased density of spines type: “thin”, “stubby” and “mushroom”. The absence of intestinal microbiota induces dendritic atrophy in other areas of the CNS, as is the case of hippocampal pyramidal neurons and granular cells of the dentate gyrus. In GF-animals, there is a significant loss of “stubby” and “mushroom” spines in hippocampal pyramidal neurons [94].

It has been estimated that there are 32% fewer synaptic connections in hippocampal pyramidal neurons of GF-animals when the dendrite size decreases and this is combined with a smaller size in the same dendritic spines [94].

A characteristic shared by the animal models of autism and GF-mice is an important alteration in the processes of social behavior. This type of alterations is in turn associated with alterations in the volume of the hippocampus and the amygdala. Changes in the size of these structures have been well documented in experiments with rodents, subject to severe stressors. Prenatally stressed rats experience an increase in the volume of the lateral amygdala [97,98] whereas chronic stress or treatment with corticosteroids induces hippocampal atrophy [98]. Changes in these structures of the CNS are frequently observed in human patients with anxiety disorders or with autism, clearly indicating that the volumetric alterations of the limbic structures can in turn be the result of a maladaptive response to stress [94]. In chronically stressed mice, dendrite hypertrophy is observed in inhibitory GABAergic neurons of the prefrontal cortex area [99].

The amygdala has different “target” areas that are responsible for modulating neuroendocrine responses to stress. The BLA area is activated by psychological stressors, and lesions in this area significantly reduce the HPA-axis response efficiency [94]. While, on the other hand, the area of the central nucleus of the amygdala (CeA) is not involved in the signaling of the HPA-axis induced by stressors, it is an area that also regulates autonomic responses to stress [94]. GF-mice have a lower degree of anxiety and social cognitive deficit, and it has been mentioned previously that there is an important relationship between anxiety and social behavior; the amygdala and the area of the ventral hypothalamus are directly involved in the regulation of this type of behavior [100]. In addition to having a preponderant role in the regulation of anxiety, the ventral hypothalamus is also involved in processes of sociability, and an alteration or damage in this area leads to the appearance of abnormal responses to social situations [101]. Besides, this ventral hippocampus exhibits a very important reciprocal connection with the amygdala, another area involved in anxiety and sociability [100].

The different tonsillar sub-regions have different roles in the regulation of anxiety and social behavior. The areas of the lateral amygdala (LA) and the BLA area integrate sensory information and adverse situations and then send their projections to the CeA area [100]. The stimulation of the projections from the BLA to the CeA area induces an anxiolytic phenotype in mice [102]. This is in contrast to direct stimulation of the entire BLA area, where an opposite effect is generated, suggesting that most of the BLA neurons project towards areas that regulate anxiogenic effects [102].

It has been mentioned that chronic stress in the adult stage is also capable of affecting the composition of the gut microbiota [11]. It is clear that alterations in the brain-gut axis interactions are associated with intestinal inflammatory processes, syndromes of chronic abdominal pain, and with eating disorders [11,103]. This altered modulation of the brain-gut axis is associated with alterations in the regulation of stress responses and behavioral alterations. The high co-morbidity that exists between stress and some symptoms of psychiatric illnesses such as high anxiety, gastrointestinal disorders (included in irritable bowel syndrome, IBS) is clear evidence of the importance of this axis in the pathophysiology of certain types of diseases [11].

Chronic stress on the other hand breaks the intestinal barrier, causes filtrations and alters the ability of the HPA-axis to reverse the deleterious effects of stress (Figure 5) [93,94].

GABA is the major inhibitory neurotransmitter in the CNS. Dysfunctions in GABA signaling are associated with anxiety and depression [104]. Lactobacilli and bifidobacteria are able to metabolize glutamate to produce GABA *in vitro* [62,104,105]. In an *in vivo* experiment in mice, a strain of *Lactobacillus rhamnosus* shows an effect and influence on depressive and ancestral behavior, and it can also alter the central expression of GABA receptors in key brain regions for stress management [62].

In 2006, Kamiya et al. [106] demonstrated that oral administration of *Lactobacillus* species for anesthetized rats is capable of completely suppressing colonic distension induced by pseudo-affective cardiac responses, which is reflected in the inhibition of visceral pain perception. This treatment is also effective in reducing electrical charges in fibers of the dorsal root of the ganglia [71]. The administration of these same strains of *Lactobacillus* to healthy adult rats is enough to activate calcium (Ca^2+^) and potassium (K^+^) channels in neurons-AH (after hyperpolarization) of the ENS in mesenteric plexus of the colon [71].

It has been shown that the oral administration of specific strains of *Lactobacillus* induces the expression of opioids-μ receptors and cannabinoids and promotes analgesic functions similar to effects of morphine. This suggests that intestinal microbiota can influence our visceral perception [107]. Altogether, these findings indicate that probiotics are able to modulate the function responsible for the visceral and somatic perception of pain [71].

Currently, there is evidence that supports the influence of intestinal microbiota on the behavior and health of SNC [1]. Patients with depressive symptoms show a significant improvement in the symptoms of depression accompanied by a reduction in plasma TRP after a fructose-restricted diet. Furthermore, fructose malabsorption provides the substrate for a rapid bacterial fermentation, which results in changes in gut motility [72]. The administration of a strain of *Bifidobacterium infantis* for 14 days increases the levels of plasma TRP, suggesting that commensal bacteria have the ability to influence the metabolism of TRP [93].

Intestinal bacteria are potent regulators of systemic and local immune responses such as that related to mucous membranes, in addition to contributing to the development of inflammatory disorders in the CNS. GF-animals or animals treated with antibiotics with an experimental autoimmune encephalomyelitis (EAE) process present reduced inflammation and a lower degree of disease compared to conventional mice, which suggest the existence of complex interactions between commensal bacteria and the inflammatory process in CNS [9,97,98]. For example, segmented filamentous bacteria (frequently associated with the intestinal epithelium) promote the development of Th17 helper T cells, which produce IL-17. They have been termed as Th17 cells in the small intestine of mice [99,108].

There is important evidence that the brain-gut axis can influence brain chemistry and is able to modulate behavior in adult mice [43]. A transient disturbance in the microbiota is able to increase the levels of BDNF in the hippocampus, as well as increase the exploratory behavior of animals. In the hippocampus, BDNF is associated with memory and learning processes and recent evidence indicates that this increase is associated with anxiolytic and antidepressant-like behavior [43]. On the other hand, the amygdala is also associated with memory and disorders in the mood and there has been an increase in the expression of BDNF in the amygdala during processes of “learning fear” [109]. Low levels of BDNF in the amygdala increase the exploratory behavior of the animals (Figure 6) [9,43].

Some other molecules with psychobiotic potential are SCFAs. These are macronutrients from non-digestible metabolites e.g., microbiome secondary degradation products of plant polysaccharides, and their production and release can be enhanced by prebiotic consumption [110,111]. These SCFAs include butyrate, acetate and propionate. It has been shown that butyrate crosses the blood-brain barrier and exhibits important neuroprotective, cognitive and anti-depressive effects [112]. Some mechanisms related to SCFAs include epigenomic histone-deacetylase gene expression regulation and HPA axis regulation [79,113,114].

Other secondary products of the psychobiotic-mediated metabolism of non-digestible fiber is DA and NE from bacilli, GABA from bifidobacteria, serotonin from enteroccocci and streptoccocci, NE and serotonin from *E. coli* and acetylcholine from lactobacilli. It is not entirely clear how much these neurotransmitters modulate the synaptic activity of the ENS [115,116,117].

The SCFAs regulate the metabolism of free fatty acids, glucose and cholesterol through various signaling cascades involving receptors linked to G-proteins [1,48]. It has also been found that acetylation of histones and SCFAs can improve cognitive function in animal models of neurodevelopment and neurodegenerative diseases, however, another group of researchers showed that the administration of a specific SCFAs, the propionic acid (PPA), can induce altered behavior traits in patients with ASD in addition to neurochemical changes [118]. These changes include neuroinflammation, elevation in levels of oxidative stress, and an important depletion in the efficiency of the antioxidant system; and all together can cause mitochondrial dysfunction, which is common in patients with ASD and in other neurodegenerative diseases such as AD and PD [119,120]. The PPA also exhibits neurotransmitter effects, effects on tight junctions and on immune function. SCFAs are associated with high levels of phosphorylated cAMP response element-binding (CREB), which induces a significant increase in catecholamine levels [121].

A first experimental study developed in rats orally administered with FOS or GOS for 5 weeks showed that both prebiotics augmented the amount of hippocampal BDNF and NR1 subunit of glutamate *N*-methyl-d-aspartate receptor (NMDAR) [122]. Besides, oral administration of GOS induced an increase in NR2A subunit expression in hippocampus, NR1 subunit and D-serine expression in frontal cortex, and plasma D-alanine. Brain levels of other amino acids related with glutamate neurotransmission were not modified by either prebiotic. Authors demonstrated that both prebiotics increased the number of fecal *Bifidobacteria*, with the effect being greater with GOS intake. However, OG may be modulating brain chemistry independently of its prebiotic activity, as gut hormones such as peptide YY (PYY) were increased in plasma of GOS-fed rats in relation with BDNF increase, suggesting a direct interaction between GOS and gut mucosa that may inclusive influence the immune system. Based in these experimental results, the same research group developed a clinical study in healthy human volunteers that received FOS, Bimuno^®^ GOS (B-GOS) or placebo daily during 3 weeks [123]. Although no effects in cortisol secretion and emotional processing were observed in relation to FOS consumption, the intake of B-GOS decreased salivary cortisol awakening response and attentional bias in participants as compared to those receiving placebo. Recently, Burokas and co-workers developed a protocol in mice supplemented with FOS, GOS or a FOS-GOS combination during 3 weeks to analyze endocrine response to stress, neurotransmitters and their receptor brain expression, gut microbiota composition, and SCFAs levels [124]. FOS-GOS treatment exhibited both antidepressant and anxiolytic effects and reduced stress-induced corticosterone release. The same decrease on corticosterone level was achieved with GOS intake, however FOS had no effect. Prebiotics also modified specific gene expression of neurotransmitters and involved-receptors in hippocampus and hypothalamus. Notably, cecal acetate and propionate concentrations were increased and that of isobutyrate was diminished by prebiotics, changes that correlated significantly with the positive effects seen on behavior. When FOS-GOS-treated mice were exposed to chronic psychosocial stress, elevations in corticosterone and proinflammatory cytokine levels, and depression- and anxiety-like behavior were reduced, as well as changes on microbiota were normalized. Thus, as previously demonstrated with probiotics, specific prebiotics may also modulate HPA axis activity and attention to emotional stimuli, suggesting a beneficial role of prebiotic treatment for stress-related behaviors.

Another path through which microbiota is able to affect functions in the CNS is by the alteration of hippocampal neurogenesis (AHN) in adults. Indeed, it has been described that the adult brain has the capacity to generate new neurons within the hippocampus and the lateral ventricles [125]. AHN is involved in memory and learning processes and can be affected by an important variety of neurological disorders such as epilepsy, major depression, AD and PD, among others [126,127]. A decrease in the number of neural stem cells and in the AHN process is observed in old age, with the concomitant cognitive decline [127]. Since metabolic and immune system pathways are involved in this process, dysbiosis of the intestinal microbiota due to diseases in the early stages of development may have long-term effects on behavior and cognitive function.

During an episode of medium stress, they observed an increase in ACTH and corticosterone release in young GF-mice, compared to young conventional specific pathogen free (SPF) mice [128]. The increases in ACTH and corticosterone levels induced by stress were completely reversed in GF mice when colonized with *B. infantis*, but only partially reversed when the mice were colonized with the microbiota of SPF mice. These findings suggest that within the microbiota of SPF mice there are bacteria that contribute to the suppression of the ACTH response.

Microbiome studies in autoimmune diseases have shown important alterations in the levels of certain bacterial groups such as *Bifidobacteria* spp. and *Lactobacillus* spp., as well as elevated levels of *Clostridia* spp., *Staphilococcus* spp. and *E. coli* [129,130,131], which are capable of alter the immune response, proinflammatory cytokine (TNF-α and IL-1-β) and anti-inflammatory (IL-10) levels, and generate feedback loops of dysbiosis while altering the immune responses. Celiac disease is a chronic inflammatory bowel disease caused by an autoimmune response to gluten [129]; in patients with celiac disease, the persistence of GABA has been suggested by mediating intracortical dysfunction despite dietary restriction. This hyperexcitability can be the result of a regulation in the GABAergic inhibitory interneurons mediated in the immune system or by a cortical reorganization mediated by glutamate, an excitatory neurotransmitter, which tries to compensate for the illness of the gluten disease [132]. In celiac disease, autoreactive clones of anti-tissue transglutaminase (anti-tTG)2 and anti-tTG6 antibodies have been found in intestine and areas of the nervous system (cerebellum, pons, bone marrow and blood vessels), addition to possible injury to the integrity of the BBB by infiltration of activated Th1 cell-exposing the brain parenchyma to the action of auto-antibodies [133]. This process leads to synaptic hyper-excitation and low inhibition at the cortical level [132,133], promoting the typical neurological signs of this disease.

Another important finding made by Sudo et al. [128] was a severe reduction in BDNF expression, at mRNA and protein levels, in the cortex and hippocampus of GF-mice, compared with SPF-mice. BDNF regulates important aspects of brain activity, including mood and cognitive functions [128]. Other reports have shown the influence of gut microbiota on the development of brain responses to stress and on cognitive functions in young mice [7,128].

## 5. Conclusions and Future Research

Nowadays, we recognize the need to study the human microbiota and probiotics as a whole ecosystem to better understand the relation between microbiota and host health or disease. One of the major limitations in using psychobiotics in humans is the lack of its possible interaction with sex hormones (estrogen and/or testosterone) and its long-term effect. Preliminary findings on how probiotic treatments, called psychobiotics, may help improve your mood, decrease your anxiety, and strengthen your memory suggest that in the near future these probiotics could be prescribed to treat depression, anxiety, and other mental health issues, by using them in the form of food or supplements to alter the gut microbiome and treat psychiatric conditions.

## Figures and Tables

**Figure 1 nutrients-11-00890-f001:**
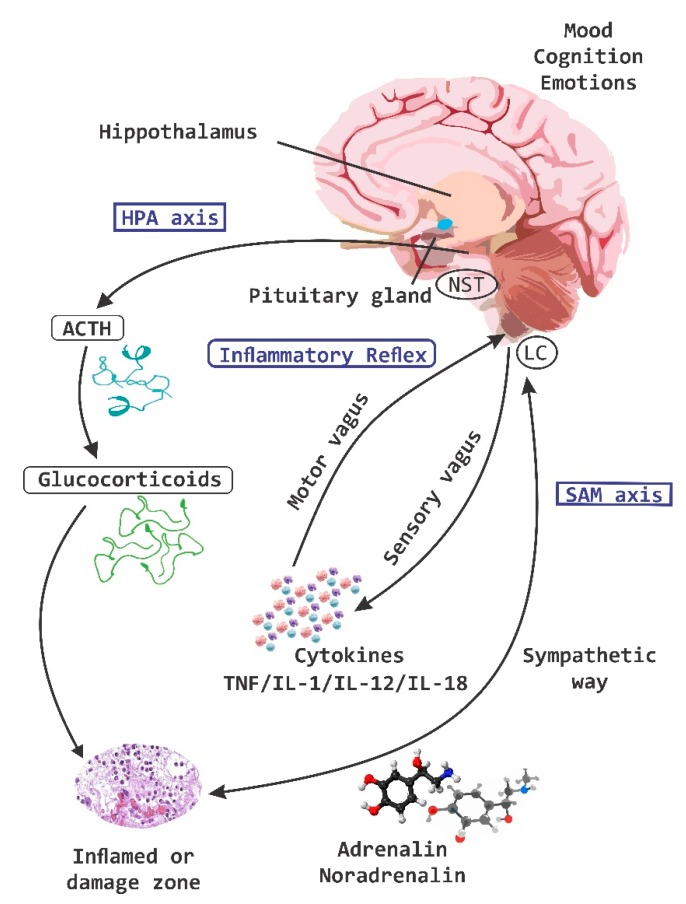
Regulation and control of neuroimmune axes. The three systems of regulation and control of information between the central nervous system (CNS) and the periphery are the hypothalamic pituitary adrenal (HPA)-axis, the sympatho-adrenal medullary (SAM)-axis and the inflammatory reflex. These systems are permanently sensing through nociceptive receptors and send information in real time to the CNS. ACTH, adrenocorticotropic hormone; NST Nucleus of the solitary tract; LC Locus coeruleus; TNF, tumor necrosis factor; IL, interleukin.

**Figure 2 nutrients-11-00890-f002:**
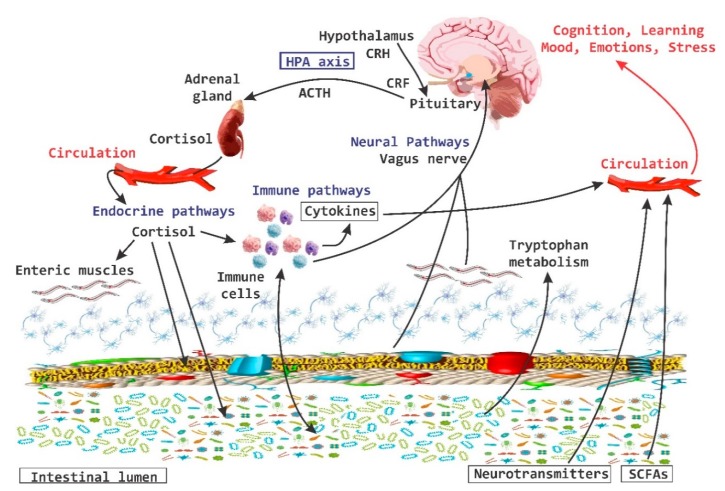
Brain-Gut Axis. The brain-gut axis is essential for the regulation established between the intestine and the brain. It includes the central nervous system and the endocrine and neuroimmune systems; as well as the enteric nervous system. CRH, corticotropin-releasing hormone; CRF, corticotropin releasing factor; SCFAs, short chain fatty acids; ACTH, adrenocorticitropic hormone; HPA, hypothalamic pituitary adrenal.

**Figure 3 nutrients-11-00890-f003:**
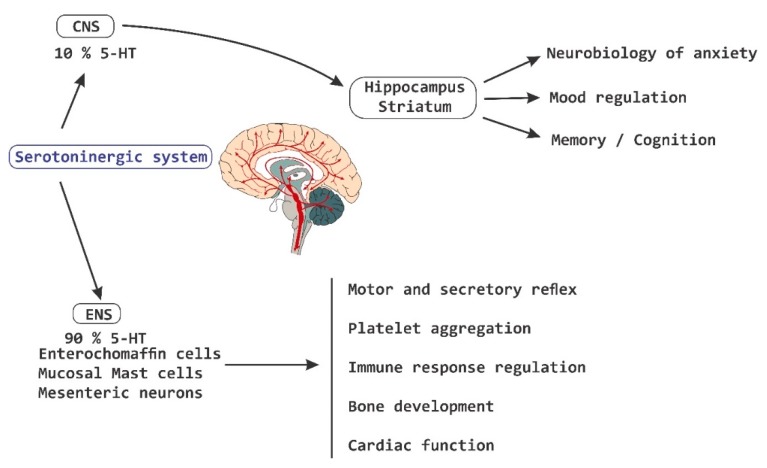
Serotoninergic system. The serotoninergic system is involved in the pathogenesis of diseases at the intestinal level, as well as in the regulation of different functions at a systemic level, which includes the regulation of memory processes, cognition and humor, among others. CNS, central nervous system; 5-HT, serotonin; ENS, enteric nervous system.

**Figure 4 nutrients-11-00890-f004:**
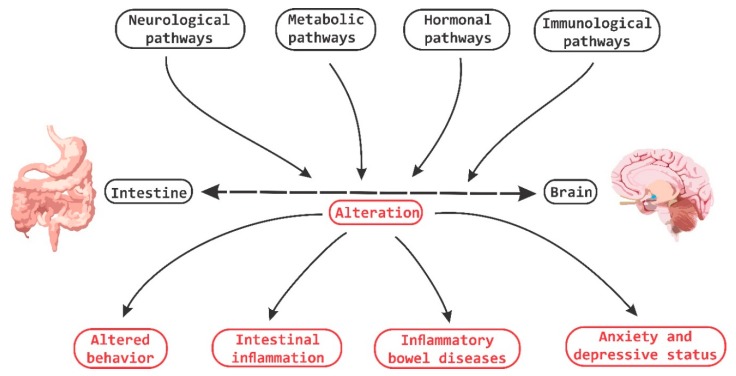
Brain-Gut Homeostasis. The relationship between the intestine and the brain involves signaling pathways at a neural, metabolic, hormonal and immune system levels. The alteration in these pathways is capable of causing changes in cognitive and behavioral processes, as well as inducing inflammatory processes at the periphery level.

**Figure 5 nutrients-11-00890-f005:**
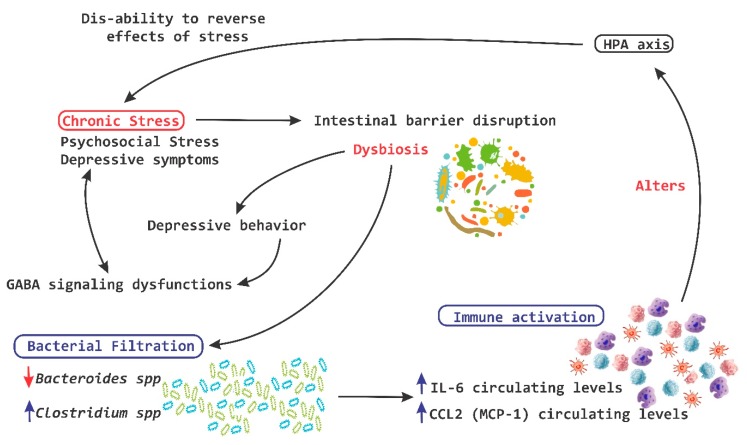
Chronic stress and HPA axis. A chronic stress process is capable of causing disruption at a level of the intestinal barrier and cause dysbiosis, which in turn induces the leakage of bacteria and the activation of the local immune system, leading to a significant alteration of the hypothalamic pituitary adrenal (HPA)-axis. IL, interleukin; MCP-1, monocyte chemoattractant protein; red arrow down mean decrease levels; blue arrow up mean increase levels.

**Figure 6 nutrients-11-00890-f006:**
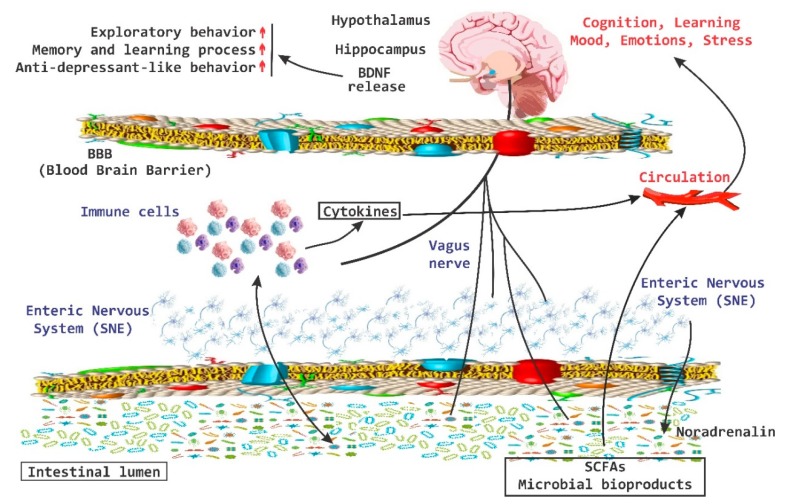
BDNF release system. The brain-derived neurotrophic factor (BDNF) released via the activation of the brain-gut axis has been associated with cognitive and behavioral processes, as well as with anxiolytic and antidepressive effects. SCFAs, short chain fatty acids; red arrow up mean increase levels.

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
