# Peer review of "From Probiotics to Psychobiotics: Live Beneficial Bacteria Which Act on the Brain-Gut Axis"

_nutrients, 2019, doi:10.3390/nu11040890_

Reviewer 1 Report

The manuscript entitled “From probiotics to psychobiotics: live beneficial bacteria which act on the brain-gut axis” is a comprehensive review of the state of the art on such an interesting, rapidly-changing field as it is the symbiont-host relationship and its effects on the neural system.

Although it is very densely written, I think this does not diminish its value. On the contrary, it provides a comprehensive review of the regulation and control of neuroimmune axes, plus an important number of citations on the interaction with the intestinal microbiota.

My main criticism is on the figures. They are interesting and helpful but quite difficult to see, with small drawings and fonts. The quality is low and, therefore, when enlarged, they get blurry and it does not help either. I think that good quality figures will greatly increase the fluency of the manuscript since it will help to non-specialists to understand the text. Sometimes it is not necessary to enlarge the overall size of the figures since there is quite a lot of empty space in them. Just increasing the font size and changing the angle of the drawing could make a difference. 

 Other minor points:

 - Figure 4: “immuinological pathways”, please correct

- Figure 5: “Depressive sumptoms”, please correct.

- Generic names of some microbial groups, like lactobacilli or bifidobacteria are written in italics sometimes and some others they are not, or they appear sometimes with a capital initial letter. Please, check through and keep the same policy through out the text. 

- The manuscript is nicely written and English is mastered but there are a few sentences that I find bizarre. These are a few examples:

 - Lines 52-53: “Many of the latter microorganisms of metabolites are known as probiotics and prebiotics respectively”

- Lines 87-88: “..panorama that changes the current paradigm of symbiosis between bacteria and humans, from this new point of view, the relationship seems to be commensalism, rather than a pure symbiosis.”

- Line 328: “capable to +infinitive”. Also in lines 349, 355, 482.

On the whole, I enjoyed reading the paper.  

Author Response

Point 1.-My main criticism is on the figures. They are interesting and helpful but quite difficult to see, with small drawings and fonts. The quality is low and, therefore, when enlarged, they get blurry and it does not help either. I think that good quality figures will greatly increase the fluency of the manuscript since it will help to non-specialists to understand the text. Sometimes it is not necessary to enlarge the overall size of the figures since there is quite a lot of empty space in them. Just increasing the font size and changing the angle of the drawing could make a difference.

Response:      The figures were edited, improving the quality, changing colors, size and type of letter for a better contrast.

They are changed in the manuscript and sent in separate files.

Point 2.-         Figure 4: “immuinological pathways”, please correct

Figure 5: “Depressive sumptoms”, please correct.

 Response:      Spelling errors were corrected in the texts of the figures.

 Point 3.-         Generic names of some microbial groups, like lactobacilli or bifidobacteria are written in italics sometimes and some others they are not, or they appear sometimes with a capital initial letter. Please, check through and keep the same policy through out the text.

 Response:      The use of italics and capitals for microbial groups is homogenized throughout the text.

 Point 4. -        The manuscript is nicely written and English is mastered but there are a few sentences that I find bizarre. These are a few examples:

- Lines 52-53: “Many of the latter microorganisms of metabolites are known as probiotics and prebiotics respectively”

- Lines 87-88: “..panorama that changes the current paradigm of symbiosis between bacteria and humans, from this new point of view, the relationship seems to be commensalism, rather than a pure symbiosis.”

 - Line 328: “capable to +infinitive”. Also in lines 349, 355, 482.

Response:      The grammatical errors of the sentences were corrected. 

Reviewer 2 Report

The authors deal with an intriguing and timely topic, which is the potential therapeutic role of bacteria acting on the brain-gut axis (i.e. psychobiotic) on some psychiatric disorders. Overall, the study is nicely conceived and designed; the results reviewed seem to be consistent and are well illustrated. However, there are some comments needing attention and revision.

- Abstract: it is too short; please include a brief background, a summary of the main findings and the translational value with clinical implications of the findings reviewed.

- Please state the aim and the rationale of the present review, highlighting the novelty compared to previous papers on this topic.

- Figure 3: “Platelet aggergation” should be changed in “Platelet aggregation”.

- Several studies reported an imbalance in the intestinal microbiota also in celiac disease (CD), thus hypothesizing that probiotics may play a role (for a recent review, see Cristofori F, et al. 2018). In order to probe the commonly observed involvement and severity of central nervous system in CD, different electrophysiological techniques have been employed (Pennisi M, et al. Front Neurosci 2017). Among them, transcranial magnetic stimulation (TMS) is a non-invasive brain stimulation technique that can contribute to the assessment and monitoring of cognition in celiac patients, even in those without a clear deficit (Lanza G, et al. Int J Mol Sci 2018). In particular, TMS in de novo patients revealed an imbalance in the excitability of cortical facilitatory and inhibitory circuits (Pennisi G, et al. PLoS One 2014). After a relatively short gluten restriction, a global increase of cortical excitability was reported, suggesting a glutamate-mediated compensation for the CD-related neurological involvement (Bella R, et al. PLoS One 2015). Conversely, after a much longer gluten-free diet, most of the electrocortical changes reverted back to normal, although some changes may persist (Pennisi M, et al. PLoS One 2017) and involve mechanisms not related to the gluten-free diet (intestinal dysbiosis? Which role for probiotics?). A comment on this aspect should be included.

- Please state the common limitations found in the studies here reviewed, as well as the pitfalls and the still controversial aspects of this topic.

Author Response

Point 1.-         Abstract: it is too short; please include a brief background, a summary of the main findings and the translational value with clinical implications of the findings reviewed.

Response:      The information requested in the summary was added.

 Point 2.-         Please state the aim and the rationale of the present review, highlighting the novelty compared to previous papers on this topic.

Response:      The request is included in the section of the summary, specifying in more detail the objective of this work and its difference with those already published

 Point3.-           Figure 3: “Platelet aggergation” should be changed in “Platelet aggregation”.

 Response:      Spelling errors were corrected in the texts of the figures.

 Point 4.-         Several studies reported an imbalance in the intestinal microbiota also in celiac disease (CD), thus hypothesizing that probiotics may play a role (for a recent review, see Cristofori F, et al. 2018). In order to probe the commonly observed involvement and severity of central nervous system in CD, different electrophysiological techniques have been employed (Pennisi M, et al. Front Neurosci 2017). Among them, transcranial magnetic stimulation (TMS) is a non-invasive brain stimulation technique that can contribute to the assessment and monitoring of cognition in celiac patients, even in those without a clear deficit (Lanza G, et al. Int J Mol Sci 2018). In particular, TMS in de novo patients revealed an imbalance in the excitability of cortical facilitatory and inhibitory circuits (Pennisi G, et al. PLoS One 2014). After a relatively short gluten restriction, a global increase of cortical excitability was reported, suggesting a glutamate-mediated compensation for the CD-related neurological involvement (Bella R, et al. PLoS One 2015). Conversely, after a much longer gluten-free diet, most of the electrocortical changes reverted back to normal, although some changes may persist (Pennisi M, et al. PLoS One 2017) and involve mechanisms not related to the gluten-free diet (intestinal dysbiosis? Which role for probiotics?). A comment on this aspect should be included.

 Response:      A paragraph concerning celiac disease at the suggestion of the reviewer added. Lines 599 - 614.

Point 5.-         Please state the common limitations found in the studies here reviewed, as well as the pitfalls and the still controversial aspects of this topic.

Response:      a paragraph was added in the conclusions section with the requested information

Round  2

Reviewer 2 Report

The authors have adequately addressed my requests, thus improving the quality of this manuscript. I do not have further comments.